# Saccades are locked to the phase of alpha oscillations during natural reading

Yali Pan[1]*, Tzvetan Popov[2], Steven Frisson[1], Ole Jensen[1]

**1** Centre for Human Brain Health, School of Psychology, University of Birmingham, Birmingham, United Kingdom, **2** Methods of Plasticity Research, Department of Psychology, University of Zurich, Zurich, Switzerland

\* yalipan666@gmail.com

**Data Availability Statement:** We have deposited the following data in the current study on figshare (https://figshare.com/projects/Pan_etal_NatCom_2021/117885): the raw MEG data, the epoch data after pre-processing, the raw EyeLink files, the

## Abstract

We saccade 3 to 5 times per second when reading. However, little is known about the neuronal mechanisms coordinating the oculomotor and visual systems during such rapid processing. Here, we ask if brain oscillations play a role in the temporal coordination of the visuomotor integration. We simultaneously acquired MEG and eye-tracking data while participants read sentences silently. Every sentence was embedded with a target word of either high or low lexical frequency. Our key finding demonstrated that saccade onsets were locked to the phase of alpha oscillations (8 to 13 Hz), and in particular, for saccades towards low frequency words. Source modelling demonstrated that the alpha oscillations to which the saccades were locked, were generated in the right-visual motor cortex (BA 7). Our findings suggest that the alpha oscillations serve to time the processing between the oculomotor and visual systems during natural reading, and that this coordination becomes more pronounced for demanding words.

## Introduction

Reading is a uniquely human skill, which relies on language comprehension as well as visual attention. During reading, we saccade around 4 times per second to place the to-be-read word in the fovea [1]. Although the visual acuity in the parafovea is relatively low, information can still be extracted [2]. Eye movement studies have found robust evidence of parafoveal processing on the sub-lexical level but have difficulties finding it on the lexical level (for a comprehensive review, please see [3]). During the intersaccadic interval that lasts around 250 ms, our brain needs to process the fixated word while also planning the next saccade. It takes at least 140 ms to identify the meaning of a visually presented word and 100 ms to initiate and execute a saccade [4]. Therefore, we argue that the oculomotor and visual systems must be intimately coordinated to support the rapid visual processing required for fluent reading.

Oscillatory activity in the alpha band (9 to 13Hz) dominates the posterior brain regions and has long been thought to reflect a brain state of rest or idling [5]. However, more recent studies show that alpha oscillations reflect the deployment of neural resources in a region-specific manner not only associated with visual processing [6–8] but also with higher-level cognitive tasks such as language processing [9]. Further work has converged on the idea that alpha oscillations might support the temporal coordination of information processing by functionally inhibiting brain regions in a phasic manner [10,11]. Recent studies found that alpha

Psychotoolbox data, and the head models after the co-registration of T1 images with the MEG data. The raw T1 images are not shared due to sensitive personal information (faces). De-identifying T1 images will remove the informative facial landmarks and make it difficult to construct head models. Therefore, we share the head models instead of the T1 images. The experiment presentation scripts (Psychtoolbox), statistics scripts (R), scripts and data to generate all figures (Matlab) are available on GitHub (https://github.com/yalipan666/Saccade-Alpha-phase).

**Funding:** James S. McDonnell Foundation Understanding Human Cognition Collaborative Award 220020448 (to O.J.), Wellcome Trust Investigator Award in Science 207550 (to O.J.), BBSRC grant BB/R018723/1 (OJ), Royal Society Wolfson Research Merit Award (to O.J.), Schweizerischer Nationalfonds zur Förderung der Wissenschaftlichen Forschung grant SNF 105314_207580 (to T.P.). The funders had no role in study design, data collection and analysis, decision to publish, or preparation of the manuscript.

**Competing interests:** The authors have declared that no competing interests exist.

**Abbreviations:** FEF, frontal eye field; HPI, head-position indicator; ICA, independent component analysis; LCMV, linearly constrained minimum variance; LIP, lateral intraparietal; PLI, phase locking index.

oscillations might be more intimately related to saccadic activity than previously appreciated [12–14]. For example, the lateralization of alpha power relates to the direction of attention-driven gaze biases during fixation [12], even in the absence of micro-saccades towards the cued location. In addition, coherent alpha activity between different recording sites in monkey V4 was found to reflect the remapping of receptive fields before and after saccades [14]. These findings point to a functional role of alpha oscillations in oculomotor control [15].

There is a need for the oculomotor system to structure the information flow within the visual hierarchy during the short-lasting fixations, in which neurons have limited time to fire. Thus, precise spiking timing is essential for optimising neuronal processing [16]. Many studies found that the phase of neural oscillations was aligned after saccade onset [17–21]. As such, the saccade onsets might modulate the excitability of the visual cortex to ensure that incoming visual stimuli arrive at the optimal phase. When monkeys freely viewed natural scenes, the saccades initiation signal modulated the V1 neuronal excitability and synchronised the visually evoked spikes afterwards on a trial-by-trial basis [17]. This saccade-related spike synchrony might result in neural amplification, i.e., fixations aligned the phase of neuronal oscillations and thus increased the impact of neural activity in the visual cortex [18]. Saccades were also found to synchronise the phase of neural oscillations downstream in the visual hierarchy, e.g., the superior temporal sulcus, when monkeys actively viewed objects [19]. The saccade-related phase resetting of ongoing oscillations may affect memory encoding. In a study where monkeys actively viewed images, hippocampal activity at 3 to 12 Hz became more synchronised after saccades when exploring images that were later remembered [20].

The phase of ongoing oscillations before saccade onset might also play an important role in visual processing. Staudigl and colleagues applied a visual exploration task and found that the phase of alpha oscillations in the occipital and medial temporal lobe aligned across trials before saccades onset. Importantly, images with stronger pre-saccadic phase-locking during exploration would be remembered during a later recall [22]. This pre-saccadic phase-locking strongly suggests that the phase of the alpha oscillations controls the initiation of eye movements during visual exploration. Likewise, the phase of alpha oscillation determines saccadic reaction times [23], the perception of near-threshold visual stimuli [24], and the awareness of visual targets [25]. This pre-stimulus phase in slow frequency oscillations was also found to play an important role in parsing continuous sensory input, determining whether to integrate or segregate 2 stimuli over time [26]. Behavioural performance of a similar temporal binding task was found to vary with phase-locking to saccades, indicating an oscillatory relationship between the saccadic eye movements and the timing of visual processing [27].

Here, we ask how the oculomotor and visual systems are coordinated to support reading. We hypothesised that the phase of ongoing alpha oscillations supports this coordination. We re-analysed a MEG dataset of a natural reading task (Fig 1A) where brain activity and eye movements were recorded simultaneously [28]. Each sentence was embedded with a low or a high lexical frequency target word. First, spectral analysis was conducted to show that the power of the alpha oscillations was strong during reading. Next, a phase-locking analysis revealed that the phase of the alpha oscillations aligned before saccade onset and was modulated by the processing demands of the parafoveal word. Finally, the neuronal sources accounting for the alpha phase-locking effect were localised to the visual cortex.

## Results

### No lexical previewing effect in fixation duration for lexicality

The eye movements from the eye tracker were parsed into events of fixations and saccades (for parameters, see Materials and methods). "First fixation duration" refers to the time of the first

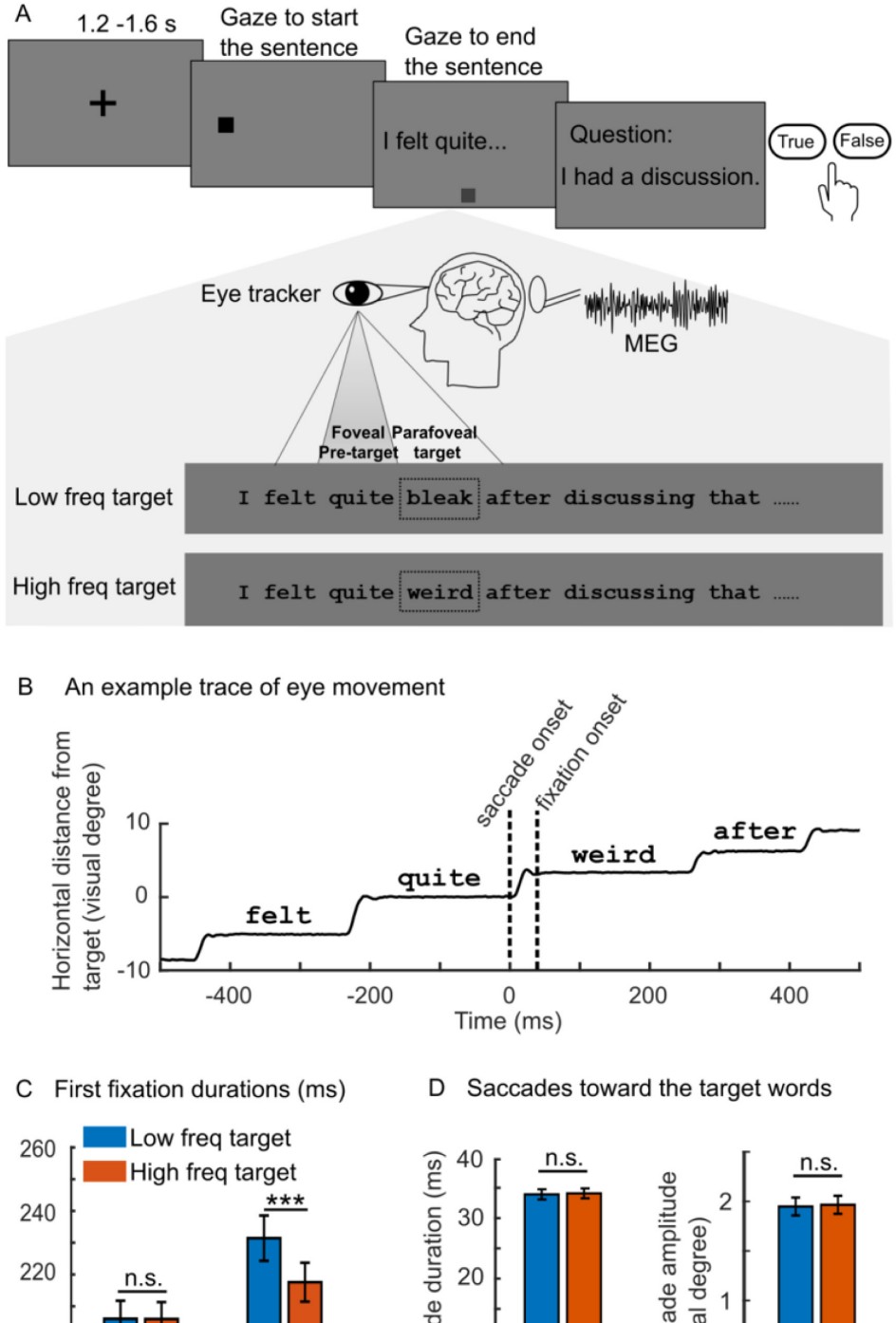

**Fig 1. The natural reading paradigm and eye movement metrics.** (A) Eye positions and brain activity were recorded simultaneously when participants ($N = 38$) read 228 one-line sentence. All sentences were plausible with unpredicted target words of either low or high lexical frequency (the dashed rectangle is for illustration only). One-quarter of the sentences were followed by a comprehension question about the just presented sentence to ensure careful reading. (B)

Horizontal eye movement trace of a given sentence was aligned with saccade onsets toward the target word "weird." The vertical dashed lines indicated saccade and fixation onset to the target word. (C) The first fixation durations were significantly longer for target words with low (blue) relative to high (red) lexical frequency (***$P = 2.1 \times 10^{-8}$, $N = 38$, two-sided pairwise $t$ test). The lexical frequency of the target word did not affect the first fixation durations of the pre-target words ($P = 0.96$, $N = 38$, two-sided pairwise $t$ test). n.s., not statistically significant. (D) Duration (left panel) and amplitude (right panel) of the saccades towards the low and high frequency target words. The data in panel (C) and (D) in the figure can be found in S1 Data.

fixation when the eye lands on the word (Fig 1B). Here, we compared the first fixation durations for pre-target and target words under the conditions of low and high frequency target words. We found a significant difference for the target words ($t_{(37)} = 7.10$, $P = 2.10 \times 10^{-8}$, $d = 2.30$, two-tailed pairwise $t$ test) but not for the pre-target words ($t_{(37)} = 0.05$, $P = 0.96$, $d = 0.02$, two-tailed pairwise $t$ test, Fig 1C). This finding demonstrates that the lexicality of the parafoveal target word does not impact the fixation durations of the pre-target word. Consistent with the literature [3,29], this suggests that eye-tracking data alone do not provide support for lexical parafoveal processing. In order to ensure that the phase modulation was not biased by any eye movement related signals, we compared the duration and amplitude of the saccades towards the target words with low and high lexical frequency. As shown in Fig 1D, no significant difference was observed for either saccade duration ($t_{(37)} = -0.77$, $P = 0.45$, $d = 0.13$, two-tailed pairwise $t$ test) or saccade amplitude ($t_{(37)} = -1.35$, $P = 0.19$, $d = 0.22$, two-tailed pairwise $t$ test).

## Alpha oscillations prevail during natural reading

We performed a time-frequency analysis for raw power aligned with saccade onsets toward the target word (Fig 2A; for the baseline corrected time-frequency representation, please see S1 Fig). The raw power was first averaged across sensors and then across all participants to obtain the grand average. We also applied the FOOOF analysis [30] to separate the aperiodic component (1/f distribution) and the oscillatory component from the power spectrum, which showed a clear peak around the alpha frequency band (Fig 2B). Fig 2C shows the topography of the alpha power in the −0.2 to 0 s pre-target interval. This 0.2 s time window was the average duration for pre-target words as observed in Fig 1C, within which the percentage of saccades showed no significant difference between conditions ($t_{(37)} = 0.94$, $P = 0.35$, pairwise $t$ test, two-sided). Alpha oscillations have long been thought to be depressed (or blocked) when processing visual stimuli [31]; however, this notion stems mainly from paradigms where visual stimuli are presented one by one on the screen. Here, we show that alpha oscillations remain strong over parietal and occipital regions when processing words during natural reading. This points to the importance of using naturalistic situations to study the functional role of alpha oscillations.

## Stronger phase-locking in the alpha band before saccades to lower lexical frequency words

The phase locking index (PLI) quantifies how consistent the oscillatory phase is over trials [32] (illustrated in S2 Fig). To uncover the role of the alpha phase during reading, we calculated PLI in the pre-target interval aligned with saccade onset to target words. Averaged duration of the pre-target interval was 0.2 s, which was used as the time window for the later permutation test. We found that the PLI in the alpha band was modulated by the lexical frequency of the parafoveal target word. Fig 3A shows the topography highlighting clusters of sensors from a permutation test supporting a significant effect ($N = 38$, $P_{\text{cluster}} = 0.006$, $d = 0.94$; see Materials and methods). Fig 3B shows the averaged PLI difference when executing saccades towards low versus high lexical frequency words over the sensor clusters in Fig 3A ($PLI_{low} - PLI_{high}$; for the raw

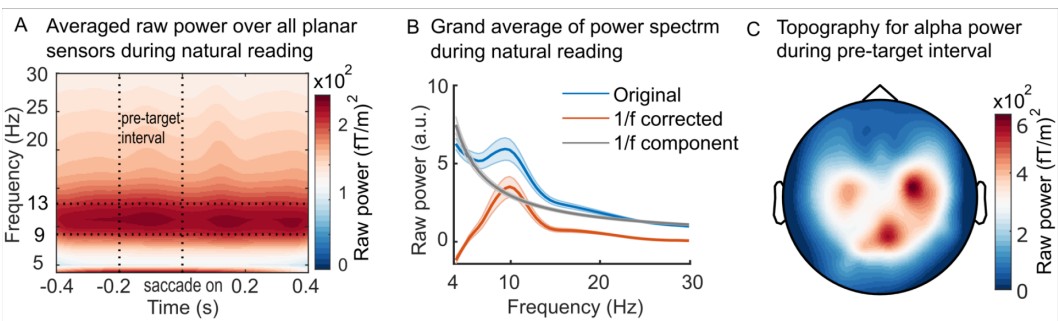

**Fig 2. Raw time frequency representation during natural reading and the topography of alpha power.** (A) Averaged power over all trials, all planar sensors, and all participants. The time-frequency representation of power was aligned with the saccade onset towards the target word. The vertical dashed lines indicate the start of the pre-target interval (−0.2 s) and the saccade onset towards the targets (0 s). The horizontal dashed lines indicate the 9–13 Hz alpha band. (B) The grand averaged power spectrum of the original data in (A), which was separated into the 1/f component (in grey) and the 1/f-correct components (in orange). The shaded areas indicate the standard error over participants (*n* = 38). (C) Topography of the averaged alpha power during the pre-target fixation interval from Fig 2A. The data in panel (A) and (B) in the figure can be found in S1 Data.

PLI for each condition please see S3 Fig). The alpha phase-locking during the pre-target interval was significantly stronger when previewing words with lower lexical frequency in the parafovea. Although the alpha phase before making saccades is more consistent over trials when previewing a low frequency compared to a high frequency word, the actual preferred alpha phase in each condition is not consistent across participants (S4B Fig). This is likely because the underlying cortical dipolar generators of the alpha activity have different orientations with respect to the MEG sensors across participants. The variability in dipole orientation results in different absolute phases of the alpha oscillations in the MEG sensors. Also, we did not find a systematic difference in absolute preferred phase when comparing saccades towards low and high frequency words (see S4F Fig). We conclude that the strength of alpha phase locking differed between conditions but not the actual preferred phase. This absence of phase-difference is consistent with a mechanism in which saccades typically are locked to the same phase of the alpha oscillations, but this locking can be stronger or weaker. Future studies based on intracranial recordings in human or nonhuman primates would allow for estimating the absolute oscillatory phase to which saccades are locked.

We conducted a control analysis to compare the PLI difference when aligned with fixation onset rather than saccade onset. The same cluster-based permutation test as in Fig 3A was performed for epochs that were aligned with fixation onset towards the target words. No robust effects were found ($N$ = 38, $P_{cluster}$ = 0.26, $d$ = 0.37) and the averaged PLI difference over the sensor cluster from Fig 3A is shown in Fig 3C. To test the specificity of the alpha frequency band, we did the same PLI analysis for the delta (1 to 3 Hz), theta (4 to 8 Hz), and beta (13 to 30 Hz) frequency bands. However, no significant difference between the low and high frequency target lexical conditions was observed (S5 Fig). Together, these results indicate the special functional role of alpha oscillations in coordinating the oculomotor and visual systems during natural reading.

Another control analysis was conducted to test whether the PLI difference found before saccades to the targets was confounded by power. For the pre-target intervals (−0.2 to 0 s aligned with saccade onset towards target words), alpha power from 9 to 13 Hz was averaged over the same sensor clusters as shown in Fig 3A. No significant alpha power difference was found between low and high lexical target conditions (S6 Fig, $t_{(37)}$ = −1.19, $P$ = 0.24, $d$ = 0.39, two-tailed pairwise *t* test).

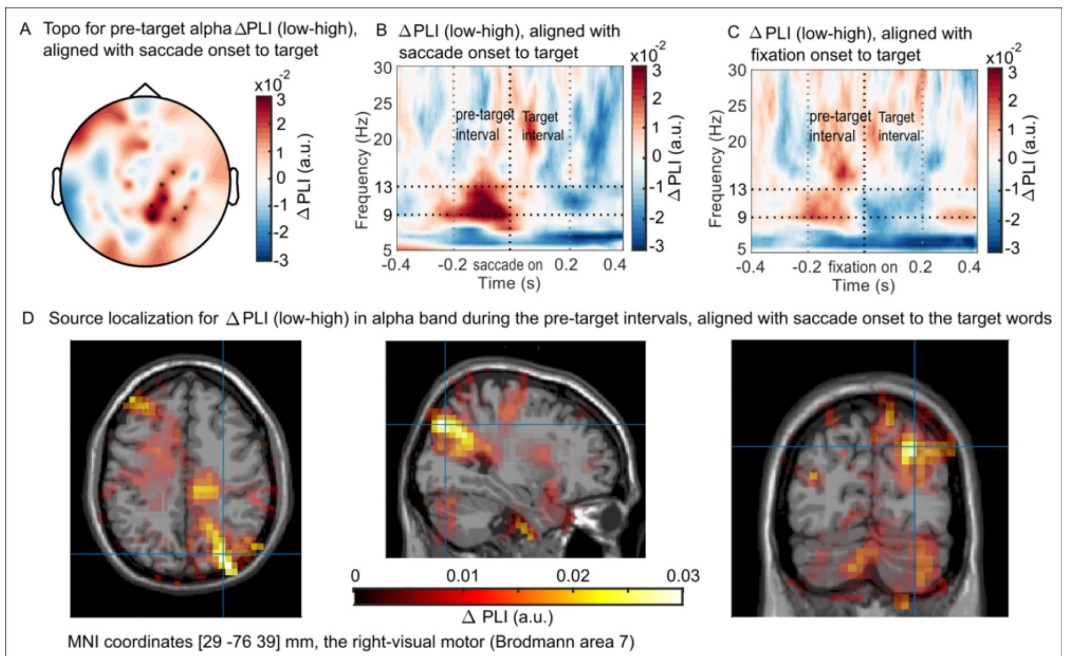

**Fig 3. PLI for the pre-target interval aligned with saccade onset to target words.** (A) Topography of the sensor clusters that showed a significant PLI difference in alpha band (9 to 13 Hz). The pre-target interval was aligned with the saccade onset towards target words (N = 38, $P_{cluster}$ = 0.006, cluster-based permutation test). (B) The averaged PLI difference ($PLI_{low}-PLI_{high}$) over the significant sensor clusters in Fig 3A. (C) Using the same sensors as in Fig 3B but aligned with fixation onset towards the target, no robust effects were observed for the PLI difference during the pre-target interval. (D) Source localization of the alpha PLI difference during the pre-target interval in Fig 3B. The generator was localised to the right-visual motor cortex (Brodmann area 7) using an LCMV beamformer. The data in panel (B) and (C) in the figure can be found in S1 Data. LCMV, linearly constrained minimum variance; PLI, phase locking index.

### The alpha PLI effect originates from the visual motor cortex (BA 7)

We used a beamformer technique, linearly constrained minimum variance (LCMV) [33], to calculate the PLI in the source space in order to localise the generators that produced the PLI initially observed in the sensors. The spatial filters were applied to the PLI difference in the alpha band (9 to 13 Hz) in the −0.2 to 0 s pre-target interval aligned with saccade onset towards target words. The parameters for the frequency band and time window were the same as for the analysis in the sensor space. The sources of the alpha-band PLI difference ($PLI_{low}-PLI_{high}$) were identified in the right-visual motor cortex (Brodmann area 7, Fig 3D).

## Discussion

We recorded the ongoing brain activity using MEG in participants reading sentences. Our core finding was that saccades were locked to the phase of ongoing oscillations. This effect was particularly pronounced when saccades were made to low compared to high frequency words. The neuronal sources accounting for this effect were localised in BA7.

### Alpha phase coordinates saccades during natural reading

To understand these findings, we first point to previous research that addressed how the inhibitory alpha oscillations might coordinate information sampling. Alpha oscillation reflects rhythmic pulses that occur around 10 times per second [34]. During each alpha cycle, the peak exerts the strongest inhibition of neuronal firing and the trough exerts the weakest inhibition; as such neuronal excitability is modulated by the alpha phase [35–38]. Given this "clocking" by

the alpha rhythm, visual processing would benefit from coordinating saccades with the alpha phase. As for natural reading, saccade timing in relation to the phase of the alpha oscillations might be important for the inflow of visual input. When the to-be-saccaded word is more difficult to process (i.e., lower lexical frequency), the timing of the saccades needs to be optimised in relation to the alpha phase. This is made possible by more strongly locking the saccades to the ongoing alpha phase when the upcoming words are rarer (low lexical frequency), as shown in Fig 3. A related effect has been found during free viewing of images: stronger alpha phase-locking before saccades during exploration results in stronger memory encoding [22] (for a systematic review, see [39]). These findings are consistent with the notion that eye movement related signals modulate the ongoing oscillatory activity in the visual system to ensure better processing of the post-fixational stimuli [40]. In the current study, we extended this notion to the language domain by studying natural reading. Together, these findings from processing images and language suggest a general neural mechanism supporting the coordination between visual and oculomotor systems across different cognitive domains. However, please note that the modulation of alpha phase-locking before saccades is based on an association analysis rather than a causal analysis. Further studies that involve causal manipulation, such as brain stimulation, are needed to establish a more causal role of phase effects of alpha oscillations on the processing of lexical information or reading behaviour in general. In the past, it has been shown that theta burst TMS to the frontal eye fields (FEFs) impair the ability to modulate alpha oscillations by spatial attention [41]. A similar approach could be applied to causally establish the role of alpha band oscillations for visuomotor coordination during reading.

In the current study, the interaction between saccades and the alpha phase was localised to the right-visual motor cortex (Brodmann area 7, Fig 3D). This result indicates that the phase modulation originated from the dorsal aspect of the parietal region, which is of great functional significance to both oculomotor control and natural reading. First, the parietal cortex is part of the oculomotor control network that involves the FEFs, motor, and visual cortex [42]. Nonhuman primates' studies found that the lateral intraparietal (LIP) area codes the motor coordinates of saccades [43] and is involved in decisions about when to initiate the saccade and where to look as well as where not to look [44]. Second, the dorsal stream of the parietal cortex is believed to be implicated in visuospatial attention [45] and damage in the posterior parietal cortex results in the core deficit of dyslexia [46]. Third, human electrophysiological studies found that alpha oscillations play a functional role within the oculomotor control network [41,47], linking oculomotor action to cognition [15]. Therefore, in the current study, alpha oscillation from the dorsal pathway might serve to coordinate the communication within the oculomotor control network, which is modulated by the previewed information during reading. Likewise, no significant difference was detected for the pre-target eye movements in relation to the lexical frequency of the target words (Fig 1C). Thus, the previewed information from the parafovea prepares the visual system in advance to ensure better processing later. In addition, this pre-saccadic phase modulation was not driven by power changes, as suggested by the null finding of the pre-target alpha power difference between conditions (S6 Fig).

Although during natural reading, saccades occur in the frequency range of the theta band, we would like to stress that saccadic patterns are not oscillatory per se. Fixation times are determined by the properties of the fixated words, such as its lexical frequency. We have recently put forward a theoretical framework making explicit how alpha oscillations might serve to coordinate visual processing during visual exploration and reading [48]. Essentially, the alpha oscillations serve to organise a pipelining mechanism operating to efficiently coordinate the different stages of word processing through the hierarchy in the ventral visual stream.

Taken together, our study provides evidence that the alpha phase serves to clock information sampling by timing the saccades during natural reading. This finding is consistent with

the idea that alpha oscillations support an oscillatory pipeline for the processing of foveal and parafoveal words during reading [48].

## Previewed lexical information from the parafovea modulates pre-saccadic brain activity in the fovea

Numerous studies have failed to provide eye movement-based evidence for lexical parafoveal processing. Specifically, fixation durations of the foveal word are not modulated by the word frequency of the parafoveal word [49–52]. Likewise, we did not find evidence for lexical parafoveal processing from the eye movement data alone. However, we did find that the consistency of the pre-saccadic alpha phase was modulated by the lexical frequency of the parafoveal word (Fig 3). This indicates that the lexical information from the parafovea affected the neural activity when processing the current words in the fovea (for further evidence please see [28]).

## Alpha power gates attention during natural reading

It has been proposed that natural reading relies on the fast modulation of spatial visual attention, but how do these processes relate to neuronal oscillations? The hemispheric lateralization of alpha activity has been thought to index the deployment of visual attention [6,53]. In the scenario of reading sentences, visual attention shifts continuously through saccades, which was supported by the fact that the alpha activity remained strong during reading (Fig 2A). Similar alpha activity and hemispheric lateralization have been found in another natural reading task where eye movements were allowed [54]. Therefore, the stage is now set to further investigate how alpha activity during natural reading shapes the visual information flow in relation to hemisphere power modulation and saccades. In a recent MEG study, Acunzo and Melcher distinguished the neural mechanisms underlying visual and non-visual processing around saccades, where occipital alpha power changes were related to the visually evoked responses [55]. Thus, the observed strong alpha power during natural reading may indicate the visual-driven responses of word processing rather than the non-visual-driven responses of oculomotor. Therefore, as in the visuospatial attention studies, alpha activity during natural reading plays an important role in shaping the visual information flow through saccades. Our results point to the importance of investigating visual processing in naturalistic scenarios rather than using a passive viewing paradigm. Now the methodological problems of doing so can be overcome [56] and naturalistic experiments have huge potential to reconceptualize the nature of human visual processing both on the level of behaviour and neural mechanisms [57].

## Active vision from saccades during natural reading

Why was the alpha phase aligned with saccade onset rather than fixation onset towards the target words? First, this is consistent with other work showing that the phase of ongoing neural activity is more related to saccade onset rather than fixation onset [17,58]. Second, saccade-related phase synchronisation might provide a more precise temporal template for active vision (for a detailed review see [59]). Since saccade initiation occurs before the visual stimuli land on our retina, the timing of neural activity in the visual cortex can be optimised to maximise the processing of the post-saccadic information. This predictive adjustment of saccadic timing is also in line with our finding that saccades were locked stronger to the alpha phase when previewing a word of lower lexical frequency. Please note that the detection of saccade onsets is better defined and relatively easier than the detection of saccade offsets (i.e., fixation onsets). This could somehow bias the estimation of the phase-locking index to be stronger for saccade onsets than fixation onsets. Third, saccadic eye movements partition the visual information flow into discrete chunks, which is consistent with the notion of active sensing [60].

Natural reading requires active sampling of visual input by saccades, which relies heavily on the interaction between the oculomotor and visual systems. Good coordination between these 2 systems enables fluent reading.

Our study highlights the importance to study "perception from action" [61]. Passive viewing paradigms are dominating the research field on vision. However, emerging evidence suggests that neural mechanisms underlying active vision are fundamentally different from passive vision [59,62–64]. Furthermore, studies that involved eye movements found that saccades had a functional role in memory formation [20,22,39], social stimuli processing [65], and temporal binding [26,27]. The stage is now set to further uncover the role of alpha oscillations in reading. In particular, it would be important to uncover how the alpha oscillations might be controlling the flow of information through the ventral stream. It would also be important to uncover the exact mechanisms by which the neuronal alpha generators interact with the saccadic motor system.

## Materials and methods

### Experimental design

We used a natural reading task (Fig 1A), where participants ($N$ = 38) read 228 one-line sentences silently. The experimental design here was within-subject, participants read an equal number of sentences containing low and high frequency target words randomly.

The background colour of the presentation screen was middle-grey (RGB = [128, 128,128]). For each trial, first, a fixation cross was presented in the centre of the screen and participants were instructed to gaze at the cross. The duration for the cross was randomly selected from a uniform distribution of 1.2 to 1.6 s. Then, a 0.5 visual degree long, white-coloured square showed up 2 visual degrees to the right of the middle of the screen's left edge. A gaze that lasted for at least 0.2 s on this square triggered the onset of sentence presentation. The accepted gaze range on the square was 1 visual degree from the square centre. Every sentence was presented in the vertical centre on the grey screen and horizontally started from 2 visual degrees to the left edge of the screen. Words were presented in black with an equal-spaced Courier New font of the size 22, which occupied 0.35 visual degrees at a distance between participants and the screen as 145 cm. The horizontal range for sentence presentation was 27 visual degrees. After participants read the sentence from the left to right silently, another gaze of at least 0.2 s on a white square 3 visual degrees below the screen centre triggered the offset of the sentence presentation. Randomly, one-fourth of the sentences were followed by a comprehension question that asked for simple facts about the just-read sentence. Participants made button presses to answer yes or no. Since all participants showed high accuracy for these questions (95.4% ± 4.7%, mean ± SD), which indicates a good comprehension of all the sentences, all trials were included in further analyses. The sentences were divided into 7 blocks almost evenly. Each lasted around 7 min with a rest of at least 1 min afterwards. The whole study took no longer than 70 min for all participants. The study was approved by the University of Birmingham Ethics Committee.

### Participants

We recruited 43 native speakers of British English (28 females), 18 to 35 years old (22 ± 2.6, mean ± SD), right-handed, with normal or corrected-to-normal vision, and without any reading disorders or neurological history. Five participants were excluded from the analyses due to falling asleep during the recordings ($N$ = 2), poor eye tracking ($N$ = 2), or too few trials ($N$ = 1), which left us with 38 participants (24 females). All participants signed an informed consent form and received either money or course credits for their participation.

## Stimuli

We used 228 sentences from 2 sentence sets (see S1 Table for the characteristics of words). Each sentence contained 1 or 2 target words somewhere in the middle. The lexical frequency of the target words was measured in terms of the total CELEX frequency per million [66]. Low lexical frequency words were defined as those with a CELEX frequency lower than 10 and the high lexical frequency words as higher than 30. Word length of the target words within a pair was kept the same. We conducted 2 pre-tests to make sure that the sentences were plausible and that participants did not predict the target words (see S1 Text for details of the pre-tests).

**Two sentence sets.** Stimuli were created using Psychotoolbox-3 [67] and were presented by a PROPixx DLP LED projector (VPixx Technologies, Canada). The refresh rate of this projector was as high as 1,440 Hz to achieve the mode of rapid frequency invisible frequency tagging, which was not the focus of analysis here (for analysis details of the rapid invisible frequency tagging of this dataset, please see [28]).

We constructed the first sentence set and got the second set from one published study [52] after removing sentences that contained the same pre-target or target words as in the first set. For a full list of sentences used in this study, please see Supplementary Information in [28]. We carried out 2 behavioural pre-tests for the first sentence set to make sure that all sentences were plausible with unpredictable target words. For the pre-test results for the second sentence set, please see [52].

The first sentence set consisted of 142 sentences with 71 pairs of target words. The sentence structure of pre-target, target, and post-target words was adjective, noun, and verb. For each pair, the target words were of the same word length but with opposite lexical frequency. For example, the pair of waltz/music, where waltz was the low lexical target and music was the high lexical target. Each target pair was embedded into 2 different but exchangeable sentence frames, and each participant read 1 version of the sentences. For the example shown below, 1 participant read version A, while another participant read version B (the target words are in bold for illustration purposes only).

A. Mike thought this difficult **waltz** received lots of criticism.
It was obvious that the beautiful **music** captured her attention.

B. Mike thought this difficult **music** received lots of criticism.

It was obvious that the beautiful **waltz** captured her attention.

Overall, all target pairs were read only once. To balance the sequence of sentences, we circularly shifted the first and second half of sentences in version B. For both versions, no more than 3 successive sentences were from the same condition of target lexical frequency.

The second sentence set consisted of 86 sentences, where each sentence was embedded with 2 target words of the same lexical frequency condition. For the example shown below, sentence A contained 2 target words of low lexical frequency, while sentence B contained 2 target words of high lexical frequency.

A. I felt quite **bleak** after discussing that really **risky** subject with Paul.

B. I felt quite **weird** after discussing that really **nasty** subject with Paul.

Each sentence frame was read once with either low or high lexical target words. In total, participants read 86 high lexical target words and 86 low lexical target words. The same control of sentence sequence was performed as in the first sentence set.

## Data acquisition

**Eye movement data.** An EyeLink 1000 Plus eye tracker (SR Research, Canada) was used to acquire the eye movement data continuously throughout the experiment. We placed the eye

tracker on a table in front of the participant. The height of the eye tracker was the same as the bottom edge of the projector screen. The distance between the eye tracker camera and the participants' eyes was 90 cm. The x and y position of the left eye and the pupil size were acquired at a sampling rate of 1,000 Hz. A 9-point calibration and validation test were performed twice during the experiment, one was at the beginning and the other was in the middle. The accepted eye-tracking error was 1 visual degree both horizontally and vertically. We also carried out a one-point drift checking every 3 trials to correct for the linear drifting. If the participant failed to trigger the onset of the sentence through the gaze on the starting square, the drift checking test was conducted again immediately and when necessary, participants were re-calibrated.

**MEG data.**   We used a MEGIN system to collect the neuromagnetic signals from 306 sensors, of which 204 were orthogonal planar gradiometers and 102 were magnetometers (Elekta, Finland). For the preparations, we first placed 4 head-position indicator (HPI) coils on the participant's head: 2 were on the forehead with more than 3 cm distance in between, and the other 2 were on the left and right mastoid bone behind the ears. Then, we used the Polhemus Fastrack electromagnetic digitizer system (Polhemus, United States of America) to digitise the head position by 3 fiducial anatomical markers (the nasion, left and right preauricular points). Next, we digitised the 4 HPI coils. For the final step, we digitised at least 200 points on the scalp to obtain the whole shape of the head. With the help of head digitalization, the MEG head position can be spatially co-registered with the individual structural MRI images for the source modelling analysis.

Afterwards, participants walked to a dimly lit room and sat in the MEG gantry (60 degrees upright position). The distance between the participant and the projector screen was 145 cm, which yielded 1 visual degree from 35 pixels. The MEG data were sampled at 1,000 Hz after the application of an anti-aliasing 0.1 to 330 Hz band-pass filter.

**MRI data.**   We used a 3-Tesla Siemens PRISMA scanner to acquire the structural T1 MRI images. The MRI session was carried out a few days after the MEG session. The scanning parameters were as follows: TR = 2,000 ms, TE = 2.01 ms, TI = 880 ms, flip angle = 8 degrees, FOV = 256 × 256 × 208 mm, 1 mm isotropic voxel. We acquired MRI data from only 36 participants due to the dropout of 3 participants, for whom the MNI template brain (Montreal, Quebec, 452 Canada) was used in the later source modelling analysis.

### Eye movement data analyses

**Extraction of eye movement events.**   The *parser* data type for each sample was set as gaze. According to the EyeLink 1000 User Manual, a conservative saccadic threshold is better for reading research. Thus, we set the following 3 thresholds to detect saccades onset during the experiment: the motion threshold as 0.1 degrees, the velocity threshold as 30 degrees/sec, and the acceleration threshold as 8,000 degrees/sec$^2$. These settings serve to prevent false saccade reports and reduce the number of micro-saccades and lengthen fixation durations. A fixation onset (or saccade offset) was detected if all these parameters were below the thresholds and the pupil was not missing; the latter case was defined as blinks. We disabled the fixation updates function by setting the fixation updates interval to zero so that the monitoring of the eye movements was as precise as the 1,000 Hz sampling rate of the eye tracker.

### MEG data analyses

**Pre-processing.**   First, the raw MEG data were band-pass filtered from 0.5 to 100 Hz using phase preserving, two-pass, fourth order Butterworth filters. Also, the data were detrended during the pre-processing. After excluding the bad sensors (0 to 2 sensors per participant), the data were decomposed into different components using the independent component analysis

(ICA, [68]). We set the maximum number of ICA steps to 100. The number of independent components was as many as the number of sensors. Components that were related to cardiac and oculomotor artefacts (e.g., eye blinks and saccades) were rejected. All data were then inspected on a trial-by-trial basis to remove artefacts that were not identified using the ICA rejection procedure.

Next, we parsed the fixation events from the eye tracker, which included the time points of fixation onset and offset, fixation duration, and the x and y coordinates. An eye position was considered a word-fixation if its x coordinate was in the range of the width of this word including the left space as well as the y coordinate was in the range of the height of this word including 2 visual degrees above and below. Fixations that landed on non-word positions were discarded. Fixations with durations less than 0.08 s or longer than 1 s were also discarded. The fixation events were aligned with the MEG data through the triggers that were sent to both systems simultaneously during the data acquisition (both systems had a 1,000 Hz sampling rate). In this way, every time point of the MEG data was assigned with fixations related to gazing at specific words. We constructed 2 types of epochs: segments were aligned with the saccade onset to the target words (i.e., the fixation offset of the pre-target words), or data were aligned with fixation onset on the target words. These 1-s epochs were centred according to saccade or fixation onsets, respectively. Finally, we visually inspected all epochs to discard those contaminated by excessive oculomotor or muscle artefacts. In further analyses, only planar gradiometer sensors were used.

**Time-frequency representations of power.**   All MEG data analyses were carried out using MATLAB (R2020a) and Fieldtrip toolbox (version 20200220 for all analyses except FOOOF analysis with version 20220208) [69].

We estimated the power for the epochs aligned with the saccade onset to target words. First, data were band-pass filtered from 4 to 30 Hz in steps of 1 Hz using Butterworth band-pass filters (two-pass, phase preserving, fourth order) following the applications of a hamming taper. The spectral smoothing was 0.3 times the centre frequency. For example, the smoothing of the band-pass filter for 10 Hz ranged from 7 to 13 Hz. For filtered data in each frequency band, a Hilbert transformation was performed to obtain the analytic signal, which then was used to estimate the power of the individual trial:

$$pow(j, f, t) = |m(j, f, t)|^2 \qquad (1)$$

where $j$ is the trial number, $f$ is the centre frequency, $t$ is the time point, and $m(j,f,t)$ is the complex value from Hilbert transformation. We estimated the raw power for each MEG planar sensor by averaging the power over trials. Then, we obtained the combined power for each planar sensor pair by taking the square root of the power from the horizontal and vertical gradients ($\sqrt{Pow_h^2 + Pow_v^2}$). The combined power was first averaged over all sensor pairs and then over all participants to obtain the grand average of power as shown in Fig 2.

**Phase locking index (PLI).**   PLI measures the phase consistency over trials [32] as shown in S2 Fig. We calculated the PLI for epochs that aligned with either saccade or fixation onset to target words. The epoch data were band-pass filtered from 4 to 30 Hz as for the power analysis above. For each participant, the same number of trials from both conditions entered the PLI analysis to avoid any bias from the trial number. For the condition that had more trials than the other, a random subsampling was carried out. Then, a Hilbert transformation was performed on the filtered data to obtain the time-varying analytic signal, which was to estimate the PLI:

$$PLI(f, t) = \left| \frac{1}{n} \sum_{j=1}^{n} \frac{m(j, f, t)}{|m(j, f, t)|} \right|, \qquad (2)$$

where $n$ is the number of trials, $m(j,f,t)$ is the analytic signal for trial $j$ at the frequency point $f$, and time point $t$. PLI value for a combined planar sensor pair was the square root of the sum of the PLI values from the horizontal and vertical planar sensor ($\sqrt{PLI_h^2 + PLI_v^2}$). A cluster-based permutation test was performed on the group level to find sensor clusters that showed significant PLI differences between conditions [70]. We had a strong hypothesis on the effect being in the alpha band [22], so the permutation test was constrained to 9 to 13 Hz. Because the averaged fixation duration of words in this study was around 200 ms (see Fig 1C), the time window for the permutation test was constrained to −0.2 to 0 s when epochs were aligned with saccade onset, and 0 to 0.2 s when epochs were aligned with fixation onset. During each permutation loop, the condition labels of PLI values were randomised over participants ($N = 38$). For each MEG sensor, a pairwise $t$ test was conducted for the shuffled PLI. With a threshold of $p < = 0.05$ (two-sided), clusters were constructed in the sensor space. The t-values within each cluster were summed, and the maximum t-value was selected to construct the null distribution. After 5,000 permutations, all t-values were sorted from minimum to maximum, both the 25th position ($PLI_{low}<PLI_{high}$) and 975th position ($PLI_{low}>PLI_{high}$) were set as the critical values for a significance level of 0.05 (two-sided). The multiple comparisons problem over MEG sensors was controlled by this cluster-based permutation test [70].

**Source analysis.** In order to localise the PLI difference observed in the sensor space, we used the LCMV beamformer technique [33] and reconstructed the time series of PLI in the source space. Based on the sensor level result, we constrained the source level analysis to the 9 to 13 Hz frequency band and the −0.2 to 0 s interval for the epochs aligning with the saccade onset to target words.

First, we prepared head models for each participant. The individual anatomical MRI images and MEG data were co-registered spatially through the fiducial markers from the head digitization during the MEG pre-preparations. For 3 participants who dropped out of the MRI session, the MNI standard template [71] was used. The aligned MRI data were segmented into a grid, where each voxel of the MRI data was assigned to a tissue class. The segmented MRI data were used to construct a single-shell head model, where spherical harmonic functions were used to fit the brain surface [72].

Second, we constructed leadfield matrices for each participant according to the forward model. A 5-mm spaced isotropic grid in the MNI normalised space was selected from the template folder of Fieldtrip. Then, each individual MRI data was warped to the normalised MRI data to obtain the transformation matrix. The inverse of this warp matrix was multiplied with the normalised grid to get the individual grid. The individual grid was not isotropic anymore, but of which the homologous points across participants were located at the same location in the normalised space. This approach allowed for the signals in the source space to be directly averaged across participants on the group level without any distortions.

Third, LCMV filters were built for each grid point in the source model using the epoch data for each participant. The epoch data were band-pass filtered in the 9 to 13 Hz range (phase preserving, two-pass, Butterworth filters with fourth order). Then, the covariance matrix was calculated for the −0.5 to 0 s interval of the filtered alpha band data over all trials regardless of conditions (please note that the estimation of covariance matrix over short periods of time is numerically challenging, so here we selected a longer time window than −0.2 to 0 s, although both time windows gave the nearly identical result, see S7 Fig). The common LCMV filters were created using this covariance matrix and the source model. We specified the LCMV filters to have a fixed dipole orientation at each source grid point, which was the optimal orientation that had the most variance. Thus, the source orientations did not vary over time. The covariance matrix was regularised with a 5% smoothing parameter. Next, in the sensor space,

we applied the Hilbert transformation to the 9 to 13 Hz filtered data to estimate the signals in the alpha band for each trial. The weight matrix from the common LCMV filters were applied to these band-pass filtered signals in the −0.2 to 0 s interval to estimate the signals in the source space (for methodological details, please see [73]). Finally, we applied Eq (2) to the source space data to estimate the PLI time series in the source space, which were then averaged across time to obtain the average PLI value for each condition.

**Statistical analysis.**   First fixation duration (i.e., the duration of the first fixation on a word) was used as the eye movement measure in the current study. All the Student's *t* tests in this study were two-sided and pair-wised, carried out by R (version 4.1.0) [74]. Effect size of the difference was measured with Cohen's d value [75].

## Supporting information

**S1 Data. Data used for the generation of main figures and supplementary figures.**
(XLSX)

**S1 Text. Two behavioural pre-tests of the sentences.**
(DOCX)

**S1 Table. Characteristics of words used in the current study.** *Note*. Position refers to the word location in a sentence where the pre-target, target, or post-target words is presented, with the unit of words. Word length means the number of letters in a given word, with the unit of letters. Lexical frequency is measured in terms of total CELEX frequency per million, where the frequency is lower than 10 for the low lexical frequency target words and higher than 30 for the high lexical frequency target words. All values in the table are mean values with standard deviations in the parentheses (mean ± SD).
(DOCX)

**S1 Fig. The corrected time-frequency representation during natural reading.** First, we averaged the power over all trials, all planar sensors, and all participants to obtain the grand average raw power. The "baseline power" was estimated by averaging the raw power over the whole temporal window. We then subtracted this baseline power from the raw power at each frequency. The vertical dashed lines indicate the start of the pre-target interval and the saccade onset (t = 0 s) to the target words. The data in the figure can be found in S1 Data.
(EPS)

**S2 Fig. Schematic diagram of the phase locking index analysis.** Alpha phase in the pre-target intervals (shaded in grey) were aligned with the saccade onset towards the target words. The averaged phase vector (radial lines in the circles) over all trials indicated the phase consistency.
(EPS)

**S3 Fig. Raw phase locking index for the pre-target interval aligned to saccade onset to the low and high lexical target words.** (A) Topography of the phase locking index with respect to low lexical target words averaged over the pre-target interval (−0.2 to 0 s) and the alpha frequency band (9–13 Hz). The black stars indicate the sensor from the cluster in Fig 3 (B). (B) Averaged PLI over the sensors in the cluster aligned with the onset of the saccade (zero time point) to the low lexical target words. The graph on the right is a zoom-in on the PLI from −0.2 to 0 s, 8 to 30 Hz. (C) and (D) are the same as (A) and (B) but for the condition of high lexical target words. The data in panel (B) and (D) in the figure can be found in S1 Data.
(EPS)

**S4 Fig. The absolute preferred alpha phase during the pre-target intervals.** We selected the planar sensor that showed the strongest phase-locking difference between conditions in Fig 3 (B) and plotted the phase locking index at 10 Hz before saccade onset (zero time point) to the target words with low word frequency (A) and high word frequency (C). The preferred alpha phase angles over all participants ($n = 38$) under the low and high target condition were shown in (B) and (D) separately. (E) The distribution of the preferred alpha phase angles at −0.1 s over participants for low (in blue) and high (in orange) condition. (F) Distribution of difference of the preferred alpha angle between conditions was not significantly different from a uniform distribution (Kolmogorov–Smirnov test, $p = 0.34$). The data in the figure can be found in S1 Data.
(EPS)

**S5 Fig. PLI difference between the low and high target lexical conditions on other frequency bands.** We calculated the PLI values in the delta (1–3 Hz) and theta (4–8 Hz) frequency bands during −0.2–0 s before saccades onset to the target words, as well as in the beta frequency band (13–30 Hz) during 0–0.2 s after saccades onset to the target words. However, no sensors survived the cluster-based permutation test of the conditional contrast (pairwise, two-tailed $t$ test).
(EPS)

**S6 Fig. Power difference during the pre-target interval between low and high lexical target conditions.** (A) Averaged raw power difference over the significant sensor cluster (in Fig 3A) during the pre-target interval (low–high). The epoch was aligned with saccade onset towards target words. (B) $T$ test of the alpha power difference for the −0.2–0 s pre-target interval between low (blue) and high (orange) lexical frequency target words. No significant difference was found ($P = 0.242$, $N = 38$, two-sided pairwise $t$ test). n.s., not statistically significant. The data in the figure can be found in S1 Data.
(EPS)

**S7 Fig. Source localization of the PLI effect with a covariance estimation time window of −0.2 to 0 s.** The covariance matrix in the alpha frequency band (9–13 Hz) was derived from the 0.2-s time window where the PLI difference was significant in the sensor space. Please note that the source localization result was nearly identical as in the main text using a −0.5–0-s time window. The difference of localised Brodmann areas was within the error of MEG recording.
(EPS)

## Acknowledgments

We thank Dr. Federica Degno and Prof. Simon Liversedge for sharing the second sentence set of the sentence material. We also thank Jonathan L. Winter for providing help with the MEG recordings. The computations described in this paper were performed using the University of Birmingham's BlueBEAR HPC service, which provides a High Performance Computing service to the University's research community. See http://www.birmingham.ac.uk/bear for more details.

## Author Contributions

**Conceptualization:** Yali Pan, Ole Jensen.

**Data curation:** Yali Pan.

**Formal analysis:** Yali Pan.

**Funding acquisition:** Tzvetan Popov, Ole Jensen.

**Investigation:** Yali Pan, Tzvetan Popov, Steven Frisson.

**Methodology:** Yali Pan, Tzvetan Popov, Ole Jensen.

**Project administration:** Yali Pan, Ole Jensen.

**Supervision:** Ole Jensen.

**Visualization:** Yali Pan.

**Writing – original draft:** Yali Pan, Ole Jensen.

**Writing – review & editing:** Yali Pan, Tzvetan Popov, Steven Frisson, Ole Jensen.

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
