## [Editor Report · Decision Letter 0]

27 Jun 2022

Dear Dr Pan, 

Thank you for submitting your manuscript entitled "Saccades are locked to the phase of alpha oscillations during natural reading" for consideration as a Short Reports by PLOS Biology.

Your manuscript has now been evaluated by the PLOS Biology editorial staff, as well as by an academic editor with relevant expertise, and I am writing to let you know that we would like to send your submission out for external peer review.

Once your full submission is complete, your paper will undergo a series of checks in preparation for peer review. After your manuscript has passed the checks it will be sent out for review. To provide the metadata for your submission, please Login to Editorial Manager (https://www.editorialmanager.com/pbiology) within two working days, i.e. by Jun 29 2022 11:59PM.

Kind regards,

Kris

Kris Dickson, Ph.D. (she/her)

Neurosciences Senior Editor/Section Manager

PLOS Biology

kdickson@plos.org

---

## [Decision Letter · Decision Letter 1]

1 Aug 2022

Dear Dr Pan,

Thank you for your patience while your Short Reports manuscript "Saccades are locked to the phase of alpha oscillations during natural reading" was peer-reviewed at PLOS Biology. It has now been evaluated by the PLOS Biology editors, an Academic Editor with relevant expertise, and by several independent reviewers. 

In light of the reviews, which you will find at the end of this email, we would like to invite you to revise the work to thoroughly address the reviewers' reports. We will be particularly looking to see you respond with further analyses to two of the major issues raised (i.e. Reviewer 1: The actual phase of alpha for the different word frequency conditions; Reviewer 2: possible effects in other MEG frequency bands). While recognizing that directly demonstrating causality could prove difficult, please note that we would also be looking to see whether your revisions convince these reviewers that the overall findings now provide the extent of conceptual advance suitable for PLOS Biology.

Given the extent of revision needed, we cannot make a decision about publication until we have seen the revised manuscript and your response to the reviewers' comments. Your revised manuscript is likely to be sent for further evaluation by Reviewers 1 and 2 as well.

**IMPORTANT - SUBMITTING YOUR REVISION**

*Re-submission Checklist*

*Published Peer Review*

*PLOS Data Policy*

*Blot and Gel Data Policy*

Sincerely,

Kris

Kris Dickson, Ph.D. (she/her)

Neurosciences Senior Editor/Section Manager

PLOS Biology

kdickson@plos.org

REVIEWS:

Reviewer's Responses to Questions

PLOS authors have the option to publish the peer review history of their article (what does this mean?). If published, this will include your full peer review and any attached files.

Reviewer #1: No

Reviewer #2: No

Reviewer #3: Yes: Benedikt Zoefel

Reviewer #1: This short manuscript reports effect of stronger phase consistency of alpha-band oscillations during natural (sentence) reading, at the onset of saccades that precede target words of great lexical complexity or lesser frequency. This report is aligned with a vein of interesting and relevant literature on the role of human alpha oscillations for modulating the inhibition of cortical circuits and/or gating the flow of information processing across brain networks. Overall, the manuscript is engaging and clearly written, with clear and relevant illustrations. However, the depth of the findings remain relatively anecdotal and may not be of sufficient interest to the broad readership of the journal. 

In my opinion, one major shortcoming of the report is that it frames the study and the discussion of the results as if the data established a causal role of phase effects of alpha oscillations on reading behavior in general, and in supporting lexical processing and/or the coordination between the visual and oculomoteur systems. Such causal demonstration is not derived in the presented work. The manuscript simply reports an association, arguably interesting in the sense that it is in line with previous observations, between the consistency of the phase of alpha oscillations across trials prior to saccading towards words and their lexical frequency. The effects are reported in terms of phase-locking across trials and compared between the two conditions of the experiment (frequent vs. less frequent words). It would be much more significant to show whether the actual phase values of alpha oscillations are different between conditions, in addition to reporting the alignment of said phase across trials. Indeed, the authors subscribe to the notion that the alpha oscillatory cycle marks time segments of lesser to stronger inhibition of cortical circuits and that stronger phase consistently between trials prior to saccading to more challenging words is a marker of the contextual modulation of such inhibitory mechanisms. However, without also exploring and reporting the consistency of the actual phase value, the paper does not entirely respond to its conceptual framework and our expectations. For instance, if less inhibition of certain visual circuits is required to process more challenging words, this should manifest with a greater aggregation of phase angle values around a certain phase segment of the alpha cycle, and the mode of such aggregation should be different between experimental conditions. 

Another issue with the results presented is with the significance of the brain region shown to demonstrate greater phase consistency in the "challenging-word" condition: MEG source mapping localizes to the dorsal aspect of the parietal region, which functional significance is uncertain. For instance, how does this region relate to oculomotor control? How is it related to parafoveal visual/reading processes that are essential in such task, in order to realign the phase of alpha oscillations. The proposed Discussion only scratches the surface of these important questions and leaves the reader with an impression that the study is incomplete and has missed opportunities to go deeper into key systems neuroscience aspects that are arguably complex, but within reach of the data.

In addition to these major issues, I have technical questions for the authors to clarify concerning MEG source mapping. The main text introduces the MEG source maps as produced from the PLI sensor values (which would be a biophysically invalid thing to do because PLI measures are statistics, not physical values). However, the Methods section mentions that the PLI values were actually derived at the source level (which is correct). Yet, these latter were obtained from projecting the Hilbert transformed sensor data into source space using an LCMV beamformer. The authors need to justify the physical validity of using the LCMV kernel, designed to project magnetic field sensor data, with the complex values of the analytical signal, which are physically meaningless. For instance, the LCMV kernel includes data covariance statistics, and I wonder how these latter differ from those of the analytical signals. Also, the Methods section mentions ITC (acronym undefined but I suppose it stands for inter-trial coherence) for deriving source PLIs, which is inconsistent. Finally, the derivation of data covariance estimates over short periods of time (200 ms) is numerically challenging (especially for alpha oscillations which cycle is 100 ms long, typically) and more detail are required beyond mentioning regularization. 

Another open question is how PLI values were derived at reach source location as no orientation constraint was used and therefore 3 time series are produced at each cortical location. 

Reviewer #2: This study investigated alpha band oscillatory activity during a natural reading task from a previous data set that manipulated the lexical frequency of a target word in a sentence. The results are consistent with a phase reset in alpha around the saccade onset time period. This reset seems stronger when the upcoming target word has low lexical frequency. These effects were localized to visual processing areas.

Overall, the finding is interesting and potentially important.

Major comment: The authors focus analyses on the 9-13 hz frequency bands. Only 4-30 Hz was considered. This means that any potential effects in the theta (or delta) bands are not considered. Given the long-standing link between saccades and theta this should be considered and explained. There is also the possibility that higher frequency effects (between 13 -30 hz) might be found, in particular in the post-saccadic interval (see Figure 3C). The authors only consider 9-13 Hz (as stated in the Methods, lines 445-446), which gives an incomplete picture given other studies in the literature showing theta or beta band effects with respect to reading and/or eye movements.

Minor comments

Line 89: Regarding alpha phase and saccades, with behavior, these studies seem relevant (both for the Introduction and even more for the Discussion:

Benedetto & Morrone (2019). Visual sensitivity and bias oscillate phase-locked to saccadic eye movements, Journal of Vision.

Wutz et al (2016). Temporal Integration Windows in Neural Processing and Perception Aligned to Saccadic Eye Movements, Current Biology. 

Line 124: "Consistent with the literature, this suggests that eye-tracking data alone do not provide support for lexical parafoveal processing." - given that the question of what information is gleaned from parafoveal processing has not been raised or at all introduced in the Introduction, this comes from nowhere within the text, is a bit complex in meaning ("eye tracking data ALONE"…), does not cite any literature supporting this statement, and perhaps does not do service to the complexity of this question within the reading literature. Given that perhaps the most interesting finding is the alpha phase locking for low-frequency words, something about parafoveal processing should be introduced earlier in the ms.

Lines 127-134 and Figure 2: It is difficult to interpret the statements about alpha "dominating" or "prevailing" (not sure about either of these choices of wording) without some context. This context is not provided until the end of the Discussion (line 227).

It is not clear if this includes some baseline or is simply a measure of oscillatory power in general, in which case one should expect 1/F power distributions (not sure if this corrects for that, as might be done with the FOOF toolbox for example). Wouldn't something like 1/F (plus perhaps some suppression of theta) give you the same pattern?

The 8-12 hz band (or 9-13 hz in this ms) is called "alpha" for a reason, in that it is the dominant pattern that can often be seen clearly even in the raw EEG/MEG trace (especially during resting state). By only looking at raw power without any baseline for comparison, it is not clear whether alpha is indeed reduced or increased relative to other conditions. I get the point, but from the analysis, it could theoretically be the case that alpha power is greatly diminished compared to some other condition. 

Also, because there is no control condition, such as saccade-only (no reading), we do not know whether the raw alpha power is driven by the saccade, by the reading task, or both. In searching for saccades and alpha, this paper looks at alpha power and phase with saccades and no task (which seems quite relevant to this ms):

Acunzo & Melcher. Changes in alpha and theta band activity at the time of saccades: separate contributions of visual and non-visual signals. PsyArXiv. https://doi.org/10.31234/osf.io/c29xm

It would seem relevant for interpreting the effects in this submission, in particular their Figures 2 and 5 showing power and phase coherence effects locked to saccade onset.

In sum, at best this figure in its present form seems to show that alpha is not eliminated during the task, which is not how it is portrayed in the text.

Lines 149-155: I found the description of phase locking to saccade onset ("the main finding") and differences in phase locking as a function of lexical frequency (the most novel finding) hard to follow. It is not clear from the text and Figure 3 which comparisons are for phase locking to saccades towards the target item in general versus for high/low frequency targets. This should be made clear, as otherwise it is hard to interpret the main findings and the abstract and first sentences of the Discussion.

p. 161: From the text and figure it seems that only the 9-13 Hz band is considered, while higher and lower frequencies are not discussed. From a quick look at Figure 3, it seems possible that PLI might also have differed in these bands (see main comment above).

Figure 3B-D: missing "…set" in "saccade onset" (not clear what "saccade on" or "saccade on to target" means otherwise).

Line 213: "This result indicates that the phase modulation originated from the visual cortex and was not biased by any eye movement related signals" - at best it "suggests" that the phase modulation was not biased by eye movement related signals. A good way to further support this idea would be to check if saccadic durations and magnitudes differed as a function of lexical frequency, since eye movement signals would tend to be correlated with the size and duration of the muscle movement.

Line 218 rewrite: "Also, this pre-saccadic modulation was purely phase-based that not

driven by the power changes…"

Line 232: "This indicates that the previewed lexical information from the parafoveal word interacts with the processing of the foveal word." A pre-saccadic difference in phase alignment does not necessarily mean that the preview interacts with the foveal processing. It could be possible to check for such effects by looking at fixation-related potentials. This issue of parafoveal previews and post-saccadic effects should at least be considered. 

Recent reviews:

Dimigen, O. & Ehinger, B.V. (2021). Regression-based analysis of combined EEG and eye-tracking data: Theory and Applications. Journal of Vision, 21, 3

Huber-Huber C, Buonocore A & Melcher D. (2021). The extrafoveal preview paradigm as a measure of predictive, active sampling in visual perception. Journal of Vision;21(7):12. doi: 10.1167/jov.21.7.12. 

Lines 235-243: This seems to mainly be conjecture. You could easily eliminate it or it would need to be re-written to reflect the complicated relationship between attention shifts and saccade onset times that make it difficult to make any strong claims based on the effects presented in this ms.

Lines 245-250: I see the point that the authors are trying to make, but this is tempered by the fact that saccade onset is a relatively well-defined moment and there is general agreement over how to measure it. Fixation onset is a completely different matter, with much less agreement over when a saccade is really complete. A simpler explanation is that saccadic onset estimates are more precise, in which case phase-locking estimates would benefit from this precision and errors in the hypothetical fixation onset would spread out these values and reduce phase locking. Indeed, even the authors spend some time in the Methods explaining their criterion for saccadic onset (lines 385-) but no details are given on how fixation onset was measured.

I did not find the argument that estimates of saccade onset are more precise INTERNALLY (in the brain) than fixation onset compelling at all. It seems to me that using efference copy to internally predict saccade onset would be relatively imprecise while the timing of re-afferent visual input is much more precise. 

Line 448: given that fixation durations in the condition of interest were matched, this is not a general problem, but it would be useful to have an estimate of how many of the the -200 to 0 intervals contained a saccade.

Reviewer #3: This study shows saccades aligned to alpha oscillations during natural reading, providing evidence for saccade timing controlled by these oscillations. This result is interesting as it extends the role of alpha oscillations as an organiser of brain activity in time to natural reading. I do not see any major issue with this work and believe that it will be well received by the community. 

I do have a few questions I hope the authors can comment on:

As the authors describe, evidence (from the same group and others) exists that saccades phase-lock to alpha oscillations in the visual system. I wonder whether the effect observed during natural reading is a specific case of this more general mechanism. In this way, the text read by the participants would be "just another" visual scene that is to be explored, and therefore dependent on alpha oscillations. Or do the authors think that natural reading is a distinct process and previous results from visual alpha oscillations do not necessarily hold?

The current focus in language/speech research seems to be on theta oscillations. Their frequency would also match that of saccades during reading (~4 Hz), suggesting that theta might be ideal to organise language processing even during reading. Could the authors speculate on why alpha is still the important frequency here? I also wonder whether results would be different if participants' actual comprehension had been tested.

The significance of "target words" for participants was not completely clear to me, did they have to detect those words?

How does the power spectrum of a typical saccade look like - is it possible that the saccadic signal biases the analysis if smeared into the relevant time intervals (by the acausal filtering performed)?

Results and methods seem to focus on the condition difference, but I think results (Fig 3B,C) for overall phase-locking (not only the difference) would also be relevant. There is a peak in the alpha band for condition difference, is it also present for overall phase-locking when tested separately for the two conditions?

---

## [Decision Letter · Decision Letter 2]

5 Dec 2022

Dear Dr Pan,

Thank you for your patience while we considered your revised manuscript "Saccades are locked to the phase of alpha oscillations during natural reading" for publication as a Short Reports at PLOS Biology. This revised version of your manuscript has been evaluated by the PLOS Biology editors, the Academic Editor and the original reviewers 1 and 3. Reviewer 2 was not available to re-review.

Based on the reviewers comments and our discussion with both the reviewer and our Academic Editor, we are likely to accept this manuscript for publication. As you will see from their comments below, the two reviewers were rather split in their views in this round. Reviewer 1 continued to raise some questions about the alpha phase differences, though during some back-and-forth discussion, Reviewer 1 did acknowledge that that one might not expect consist phase differences between conditions if phase locking is weaker for low-frequency words. However, Reviewer 1 still questioned how strongly the study can inform on our mechanistic understanding alpha and inhibition during reading absent information about specific phases. Given that our readership might have similar questions and concerns to those still being expressed by Reviewer 1, we ask that you explicitly comment on these points in the final version of the article. 

***Please also make sure to address the data and other policy-related requests listed below my signature are fully addressed. Addressing these issues is necessary for publication and incomplete responses and edits will result in publication delays.***

We expect to receive your revised manuscript within two weeks. 

*Published Peer Review History*

*Press*

Sincerely,

Kris

Kris Dickson, Ph.D., (she/her)

Neurosciences Senior Editor/Section Manager,

kdickson@plos.org,

PLOS Biology

DATA POLICY:

Thank you for depositing your raw data in figshare, complying with the PLOS Data Policy which requires that all data be made available without restriction: http://journals.plos.org/plosbiology/s/data-availability. Note that we do not require all raw data but appreciate it's inclusion in those files. 

1) Please make sure that you also provide the specific summarized data used to create your figures and make available either in the figshare link or as supplementary files (e.g. excel). This will need to be done for the following graphical displays:

Fig 1C,D; Fig 2A,B; Fig3B,C in the main paper.

Please also include all of the data necessary to reproduce the supplemental data as well.

2) Please save these summary data files with a description field using the following format verbatim: S1 Data, S2 Data, etc. Multiple panels of a single or even several figures can be included as multiple sheets in one excel file that is saved using exactly the following convention: S1_Data.xlsx (using an underscore).

3) underlying data can be found (i.e. The data in Fig X, panel Y can be found in S1 Data...).

4) Please also update your Data Statement in the submission system accurately describes where your data can be found.

DATA NOT SHOWN?

- Please note that per journal policy, we do not allow the mention of "data not shown", "personal communication", "manuscript in preparation" or other references to data that is not publicly available or contained within this manuscript. Please double check that your submission does not contain such statements and either remove mention of any such data or provide figures presenting the results and the data underlying the figure(s).

Reviewer remarks:

Reviewer's Responses to Questions

Do you want your identity to be public for this peer review?

Reviewer #1: No

Reviewer #3: Yes: Benedikt Zoefel

Reviewer #1: I appreciate the attention brought by the authors to the comments and suggestions I provided after reviewing the first version of their manuscript. 

Although the discussion of the results is now more in line with the relatively modest effects reported, I still see issues with the significance of the effects reported, their interpretation in terms of neurophysiological mechanisms, and with the methodological approach to source imaging. 

In terms of the significance of the effects, I am grateful for the efforts put in testing the consistency of the phase differences observed between the experimental conditions and across participants. It is indeed disappointing that the actual values of the alpha phase are not systematic between participants, and to a more substantial concern, that their differences between conditions within participants are not consistent. This latter is concerning because, although the absolute phase of the alpha oscillations can be difficult to detect and interpret across participants, as the authors now explained, the relative difference of the phase values should be interpretable. Indeed, if the alpha cycle marks the phases of relative greater/lesser inhibition, the phase differences between conditions should be expected to indicate a systematic change at least towards the direction of a trough or a peak of the oscillation, across participants. Without this result, I find the report less convincing and still quite far from a mechanistic and generalizable interpretation of the findings.

The explanation provided concerning the LCMV weights applied to a complex signal in lieu of the physical MEG physical measures is not convincing either unfortunately. The authors cite a 2008 paper that used a similar approach, and a Fieldtrip email thread where the argument is that "it should basically work." I don't believe this is convincing. 

The source maps have been recomputed using a longer time segment for the computation of data covariance (thank you). For completeness, the display of Figure 3D should show MR crossections in all three directions (axial, coronal and sagittal) and using a glass-brain rendering so that the reader can better appreciate the location and extension of the activated regions. 

In general, all delta_PLI effects should be reported and plotted as relative values i.e, in percentages of PLI statistics changes between conditions.It would make clearer that the effects reported are remarkably small, and based on a relatively small cohort. 

Minor:

p.8 - "The alpha phase during the pre-target interval was significantly stronger when previewing words with lower lexical frequency in the parafovea...": the alpha phase cannot be stronger. I believe this is a typo and that the authors refer to alpha phase locking.

Reviewer #3: Thank you for answering my questions on what I believe is an important study on the role of alpha oscillations for reading and visual scene analysis in general.

---

## [Editor Report · Decision Letter 3]

16 Dec 2022

Dear Dr Pan,

Thank you for the submission of your revised Short Reports "Saccades are locked to the phase of alpha oscillations during natural reading" for publication in PLOS Biology. On behalf of my colleagues and the Academic Editor, Chris Pack, I am pleased to say that we can in principle accept your manuscript for publication, provided you address any remaining formatting and reporting issues. These will be detailed in an email you should receive within 2-3 business days from our colleagues in the journal operations team; no action is required from you until then. Please note that we will not be able to formally accept your manuscript and schedule it for publication until you have completed any requested changes.

PRESS

Sincerely, 

Kris

Kris Dickson, Ph.D., (she/her)

Neurosciences Senior Editor/Section Manager

PLOS Biology

kdickson@plos.org